# Tumor Suppressor miRNA-503 Inhibits Cell Invasion in Head and Neck Cancer through the Wnt Signaling Pathway via the WNT3A/MMP Molecular Axis

**DOI:** 10.3390/ijms232415900

**Published:** 2022-12-14

**Authors:** Shang-Ju Tang, Kang-Hsing Fan, Guo-Rung You, Shiang-Fu Huang, Chung-Jan Kang, Yi-Fang Huang, Yu-Chen Huang, Joseph Tung-Chieh Chang, Ann-Joy Cheng

**Affiliations:** 1Department of Medical Biotechnology and Laboratory Science, College of Medicine, Chang Gung University, Taoyuan 33302, Taiwan; 2Graduate Institute of Biomedical Sciences, College of Medicine, Chang Gung University, Taoyuan 33302, Taiwan; 3Department of Radiation Oncology, New Taipei Municipal TuCheng Hospital, New Taipei City 236017, Taiwan; 4Department of Otorhinolaryngology—Head and Neck Surgery, Linkou Chang Gung Memorial Hospital, Taoyuan 33302, Taiwan; 5Graduate Institute of Clinical Medical Sciences, College of Medicine, Chang Gung University, Taoyuan 33302, Taiwan; 6Department of General Dentistry, Linkou Chang Gung Memorial Hospital, Taoyuan 33305, Taiwan; 7Department of Radiation Oncology and Proton Therapy Center, Linkou Chang Gung Memorial Hospital, Taoyuan 33302, Taiwan; 8School of Medicine, Chang Gung University, Taoyuan 33302, Taiwan

**Keywords:** miR-503, head and neck cancer, invasion, Wnt signaling pathway, WNT3A

## Abstract

Head and neck cancer (HNC) is the fifth most common cancer worldwide, and its incidence and death rates have been consistently high throughout the past decades. MicroRNAs (miRNAs) have recently gained significant attention because of their role in the regulation of a variety of biological processes via post-transcriptional silencing mechanisms. Previously, we determined a specific profile of miRNAs associated with HNC using a miRNA microarray analysis. Of the 23 miRNAs with highly altered expression in HNC cells, miR-503 was the most significantly downregulated miRNA. In this study, we confirmed that miR-503 acts as a tumor suppressor, as our results showed decreased levels of miR-503 in cancer cells and patients with HNC. We further characterized the role of miR-503 in the malignant functions of HNC. Although there was a minimal effect on cell growth, miR-503 was found to inhibit cellular invasion significantly. Algorithm-based studies identified multiple potential target genes and pathways associated with oncogenic mechanisms. The candidate target gene, WNT3A, was confirmed to be downregulated by miR-503 at both the mRNA and protein levels and validated by a reporter assay. Furthermore, miR-503 modulated multiple invasion-associated genes, including matrix metalloproteinases (MMPs), through the Wnt downstream signaling pathway. Overall, this study demonstrates that miR-503 suppresses HNC malignancy by inhibiting cell invasion through the Wnt signaling pathway via the WNT3A/MMP molecular axis. The modulation of miR-503 may be a novel therapeutic approach to intervene in cancer invasion.

## 1. Introduction

Cancer, also known as cell malignancy, is the second leading cause of disease-related death worldwide. With a high mortality rate, coupled with a reduction in the quality of life, cancer poses one of the greatest challenges to human health. Head and neck cancer (HNC) is one of the most common and recurring forms of cancer, with approximately 900,000 cases annually [1,2,3,4]. While HNC generally refers to cancer in the oral region, other affected areas include the nasal cavity, oral cavity, larynx, and pharynx [5].

Various treatment methods, such as surgery, chemotherapy, and radiotherapy, have moderate success rates; however, the most important factor that affects the success rate of curing cancer is not how the cancer is treated but rather when it is treated. Regardless of the cancer type, the survival rate in early-detection scenarios increases by approximately 80% compared with late-stage detection [6]. Not only would the knowledge of these carcinogenic pathways and downstream target genes be useful for the diagnosis of cancer at very early stages, but the associated molecules may also help in monitoring the prognosis as well as in evaluating the post-treatment conditions in case a relapse occurs [7]. Recently, many different pathway genes were studied and reported as promising candidates, although further validation is required [8]. These potential markers included chemokine receptors [9], cytokeratins [10], interleukins [11], metalloproteinases (MMPs) [12], and microRNAs (miRNAs) [13].

miRNAs are short, non-coding RNAs of approximately 22 nucleotides in length. They can be identified in cells and are secreted into multiple body fluids, including serum, saliva, and urine [14], and can function as oncogenes or tumor suppressors through post-transcriptional gene regulation [15]. Changes in the homeostasis of cancer-associated miRNAs are a common occurrence in both solid tumors as well as the surrounding environment due to the shift in cell phenotype [16,17]. Many miRNAs were found to be up or downregulated in multiple types of cancers, including lung [18], breast [19], and brain cancers [20]. Naturally, HNC has a fair proportion of associated miRNAs. For example, miR-155 and miR-196b were previously reported to play oncogenic roles by regulating the STAT3/SOCS1 and MMP pathways, respectively, while miR-34a and miR-125b were downregulated and acted as tumor suppressors through the regulation of RTCB and PRXL2A, respectively [21,22,23,24,25]. These carcinogenic activities resulted in uncontrolled cell proliferation, the evasion of cell death, angiogenesis, increased migration and invasion ability, and, consequently, cancer development [26,27,28]. Additionally, studies have shown that alterations in the levels of circulating miRNAs during cancer progression could be detected through liquid biopsies [29]. Together, these results suggest that miRNAs could act as quality diagnostic and prognostic tools in clinical cancer research.

Previously, our research team performed miRNA profiling and the data analysis of dysregulated miRNAs in HNC using miRNA microarrays [30]. Among the candidate miRNAs, miR-503 was found to be the most significantly downregulated miRNA. miR-503 is located on chromosome Xq26.3 and was studied in a variety of diseases, including cancer. Its most reported functions include proliferation and metastatic regulation in cancers [31]; it was also indicated in lncRNA-related studies [32]. While some reports have shown that miR-503 has oncogenic functions, others have concluded that it acts as a tumor suppressor. For example, the downregulation of miR-503 inhibited cancer growth in gastric, hepatocellular, osteosarcoma, prostate, and lung cancers [33,34,35,36,37]. A recent study by Li et al. also found that miR-503 inhibited esophageal cancer progression [38]. Conversely, the overexpression of miR-503 in colorectal and esophageal cancers resulted in tumor progression [39,40]. However, to our knowledge, the interactions between miR-503 and HNC remain unclear. While some studies have indicated that its expression is elevated in HNC [41], other profiling studies have identified miR-503 as a tumor suppressor [42,43]. Additionally, although a few downstream pathways of miR-503 were proposed, many require further verification.

In this study, we examined the function of miR-503 and found that the invasion function was predominantly associated with HNC progression. We further characterized the miR-503 regulatory mechanism and identified WNT3A as a direct target that affects its downstream molecular pathway. Our results provide evidence that miR-503 may decrease HNC invasion through the Wnt signaling pathway by targeting WNT3A.

## 2. Results

### 2.1. miRNA-503 Is Downexpressed in Patients with HNC as Well as HNC Cell Lines

To determine the relevance of miR-503 expression in HNC, we first analyzed the expression levels of miR-503 in normal and cancer cells. Seven HNC cell lines (OECM1, SAS, FaDu, SCC25, OC3, CGHNC8, and CGHNC9) and five normal oral keratinocyte cell lines (CGHNK2, CGHNK4, CGK6, CGK5, and CGK1) were examined. We found that the miR-503 expression was much lower in cancer cell lines than in normal cells (*p* = 0.018, Figure 1a). Subsequently, we compared the miR-503 levels in the plasma samples from patients with HNC and healthy individuals (Figure 1b). Similar to the in vitro examination, the miR-503 levels were significantly lower in patients with HNC than in healthy individuals (*p* = 0.0055, Figure 1b). These results suggested that miR-503 acts as a tumor suppressor in HNC.

### 2.2. miRNA-503 Has a Minimal Effect on Cell Growth but Suppresses Cell Invasion in HNC Cells

To determine the potential function of miR-503, in vitro gain-of-function experiments were performed by transfection with miR-503 mimic oligonucleotides. Three HNC cell lines (SAS, OECM1, and FaDu) were examined to obtain more general results. The effect on cell growth was examined using the colony formation assay. As shown in Figure 2a, no significant alterations in colony growth in any of the tested cell lines were observed when the cells were transfected with miR-503. Additionally, the cell proliferation assay showed no significant effects on the cell growth either, further indicating that miR-503 does not affect HNC cell growth (Appendix A). The effect on cell invasion was determined using the Matrigel invasion assay. As shown in Figure 2b, miR-503 significantly suppressed the ability of cancer cells to invade the surrounding areas; the number of invaded cells transfected with miR-503 was greatly reduced compared to the control groups by at least 45% across all three cell lines (Figure 2b, bottom). A wound-healing migration assay was also performed, though no notable effect was seen when miR-503 was upregulated in HNC cells (Appendix A). These results suggested that miR-503 mainly affects HNC progression by reducing cell invasion.

### 2.3. WNT3A Is Directly Inhibited by miRNA-503 in HNC

For searching the potential target genes of miR-503, several miRNA target prediction algorithms, including starBase, TargetScan, DIANA, miRanda, and miRSystem, were used to screen for the downstream targets. The results are shown in Figure 3a. The starBase dataset analysis identified 1403 genes, while the TargetScan, DIANA, miRanda, and miRSystem analyses identified 311, 560, 1083, and 227 genes, respectively. In total, 2281 unique genes were identified across the five databases. To further determine the potential molecular regulatory pathways, the Kyoto Encyclopedia of Genes and Genomes (KEGG) enrichment analysis of the 2281 uniquely predicted genes was carried out. As shown in Figure 3b, the KEGG analysis revealed that the most prevalent pathways, including the Wnt, MAPK, and PI3K-AKT signaling pathways, were associated with cancerous functions such as cell growth, cell motility, and Wnt signaling.

To confirm the potential and accuracy of the predicted target genes of miR-503 in HNC, eleven genes in common across the five prediction databases, including BCL11B, CCND2, EIF3E, FGF7, JARID2, PDCD4, PTK7, RICTOR, SMAD7, WNT3A, and ZNF217, were verified. These genes were analyzed using RT-qPCR, and their expression was quantified, as shown in Figure 3c. Of the eleven genes, WNT3A, BCL11B, and CCND2 were the most significantly downregulated, with all three average expression levels under 0.8-fold post-miR-503 regulation. WNT3A was found to be the most downregulated, with an average fold change of 0.62 across all cell lines (Figure 3c, bottom). Further examination revealed that the basal expression levels of WNT3A were inversely correlated with miR-503 levels in both HNC cell lines as well as normal keratinocytes, supporting our hypothesis that WNT3A may be a direct target of miR-503 (Appendix A). 

A Western blot analysis was performed to examine the protein expression. The protein analysis showed that miR-503 overexpression significantly downregulated the protein expression of WNT3A across all three cell lines (Figure 3d). Next, we confirmed the interaction between miR-503 and WNT3A using a dual-luciferase reporter assay. The binding site of miR-503 on the 3′-UTR region of WNT3A was characterized using TargetScan, and the wild-type and a mutant form of the 3′-UTR region of WNT3A were designed as shown in Figure 3e. As shown in Figure 3f, the luciferase assay revealed that the wild-type group had an approximately 40% decrease in luciferase activity in the SAS and OECM1 cell lines and a 50% decrease in the FaDu cell line in the presence of miR-503. Rescue experiments were performed to validate the role of WNT3A in HNC. The overexpression of miR-503 alongside a Wnt agonist demonstrated the restoration of the invasive phenotype when it reversed the anti-tumor effects of mir-503 in HNC, as shown in Figure 3g. Taken together, these results confirmed that WNT3A is directly regulated by miR-503 in HNC. 

### 2.4. miRNA-503 Modulates Multiple Invasion-Associated Genes, including MMPs, through the WNT3A Downstream Signaling Pathway

WNT3A is a ligand in the canonical Wnt signaling pathway. Most cancer pathways have a complicated network of genes that may affect each other. Hence, we examined Wnt signaling targets that were significantly dysregulated by HNC cells overexpressing miR-503. Using a PCR array, we screened 84 Wnt signaling-associated genes. A fold regulation (FR) greater than 2 (|FR| ≥ 2) was set as the screening threshold criterion, and 55 genes were found to be significantly dysregulated in the presence of miR-503 overexpression (Figure 4a, Appendix A). In addition, 41 of these 55 genes were downregulated, suggesting that miR-503 directly inhibited the gene expression. As expected, the expression of WNT3A was significantly downregulated. We further found that other invasion-associated genes, including MMP7, FGF7, and CTGF, were also downregulated.

Since our initial functional analysis revealed that miR-503 inhibited the invasive capabilities of HNC, we investigated the pathways and genes associated with invasion and verified the invasive capabilities of miR-503 in HNC cell lines using a molecular model. Genes, such as those of the matrix metalloproteinase (MMP) family and other mesenchymal markers, were analyzed with RT-qPCR. Notably, the expression levels of invasion-associated genes, viz. MMP3, 7, and 9, were the most commonly and significantly downregulated across the SAS, OECM1, and FaDu cell lines (Figure 4b). Other noteworthy genes included fibronectin, vimentin, and WISP1, all of which play roles in Wnt signaling, as well as have metastatic functions. Although further testing is required to confirm which MMPs have the strongest correlation with miR-503 and WNT3A, we could infer that the inhibition of WNT3A could result in a decreased MMP expression level, leading to reduced invasive capabilities. 

## 3. Discussion

Head and neck cancer is one of the most common cancers worldwide, particularly in southeast Asia. While various treatment methods are available, many have harmful side effects, and the recurrence potential is high. In this study, we investigated the downstream activities and molecular mechanisms of miRNA-503. This study highlights several key findings: (1) miR-503 was found to act as a tumor suppressor in HNC; (2) miR-503 plays an important role in HNC progression by inhibiting invasion; (3) miR-503 directly regulates WNT3A and the downstream genes of the Wnt signaling pathway in HNC; and (4) miR-503 regulates the invasive ability of HNC cells through MMP family genes.

The microarray screening performed in our previous studies allowed us to select miR-503, a novel tumor suppressor in HNC. Many studies revealed that this miRNA plays an important functional role in various cancers [44]. In general, miR-503 acts as a tumor suppressor; numerous studies on osteosarcoma and gastric, prostate, hepatocellular, and lung cancers were conducted to prove this capability [33,34,35,45,46]. In contrast, a few studies have shown that miR-503 may play an oncogenic role in other types of cancers. For example, Zhao et al. found that miR-503 induces breast cancer through epithelial-mesenchymal transition [47], and Cheng et al. found that miR-503 targets PTPN12 to promote retinoblastoma [48]. This variation in expression between different cancers is not uncommon and is most likely due to different localizations and tissue-specific factors, in addition to the ability of miRNAs to target multiple mRNAs [49]. Herein, we found that miR-503 behaved like a tumor suppressor, for it was underexpressed in both the HNC cell lines and patients with HNC (Figure 1).

Although some studies found that miR-503 inhibits cell proliferation in cervical, glioma, and gastric cancers [50,51,52], our study shows that miR-503 may not affect this particular function in HNC. Instead, the functional analysis results indicated that the function of miR-503 in HNC is less related to cell growth and more to cell invasion (Figure 2). In line with our findings, Jiang et al. and Wei et al. also found that miR-503 promotes invasion and metastasis in hepatocellular and colon cancer, respectively [53,54]. The invasion function, one of the key hallmarks of cancer, is crucial for cancer progression. It describes the invasive ability of cancer cells to migrate to distant secondary sites in the body, usually through blood vessels or lymph nodes. During this course, many processes and genes, including those from the epithelial-mesenchymal transition and remodeling of the extracellular matrix, are altered to allow cancer cells to invade the nearby vessels better [55,56]. Matrix metalloproteinases (MMPs) are the most well-known matrix-degrading enzymes that play important roles in tumor cell invasion [57,58]. As the head and neck region of the body contains many lymph nodes, the risk of metastasis is extremely high. Fundamentally, the invasion niche is the main reason for the poor prognosis and high death rate of patients with HNC. An investigation regarding whether the inhibition of invasion in HNC caused by miR-503 is through the dysregulation of extracellular matrix molecules, such as MMPs, could prove useful for prognostic and therapeutic studies.

To further confirm that this functional phenotype in HNC is caused by the dysregulation of miR-503, we examined the potential genes that may be associated with miR-503 through bioinformatics analysis. Because miRNAs can target multiple transcripts, a single miRNA may regulate a multitude of mechanisms, which may lead to varying outcomes in cancer functions [59,60,61]. A plethora of computational tools based on the seed region, conservation, free energy, and site accessibility have been developed in recent years [62], allowing the easy screening and selection of the most suitable candidates for further experimentation. Although the predicted genes may not be directly targeted by the miRNA, owing to the nature of the prediction criteria, these molecules and their functions may still be indirectly associated. Therefore, in our study, we used the following five different bioinformatics databases to predict the target genes of miR-503: TargetScan [63], miRSystem [64], DIANA [65], miRanda [66], and starBase [67]. The predicted genes were assessed using a KEGG enrichment analysis. A KEGG analysis is an integrated database for the biological interpretation of genes, which allows us to determine the functions and ontologies associated with the gene list provided for the analysis [68,69]. Because the pathways in KEGG were studied countless times across various diseases, including cancer, we found that many of the pathways could be grouped into functional categories. Specifically, the KEGG analysis indicated that miR-503 participates in cell growth and motility (Figure 3b). Cell senescence is also noted in the predicted pathway. Regarding this, our cell proliferation assay showed no significant difference in the growth rate after miR-503 modulation (Appendix A). Since a slower growth rate often accompanies cellular senescence, the insignificant effect of cell proliferation by miR-503 overexpression implies that miR-503 may have a minimal role in senescence regulation. Nevertheless, further experiments may be needed to confirm this postulation. We found the Wnt signaling pathway to be highly significant and reoccurring in multiple KEGG sets, which led us to believe that while miR-503 may play roles in different functions, the Wnt signaling pathway in particular could be important in HNC.

Our target prediction analysis also allowed us to screen for the best candidates that could directly bind to miR-503. Because each database applies slightly different prediction algorithms and formulas to its computational approach, a vastly different gene list was generated for each search. To circumvent this, we focused on mutual genes found across all five databases. Eleven candidate target genes of miR-503 were screened, and their expressions were validated with RT-qPCR (Figure 3c). Because miRNAs generally inhibit gene expression, we mainly addressed the genes that were downregulated. Of the 11 genes, the expressions of WNT3A, CCND2, and BCL11B were most significantly inhibited in the SAS, OECM1, and FaDu cell lines, with average inhibition levels of 0.52, 0.59, and 0.70, respectively. CCND2 and BCL11B are known to have functions specific to the cell cycle or growth, and T-cell regulation, respectively [70,71], implying that miR-503 may also modulate these functions, although these may not be the main functions in HNC. Nevertheless, we decided to focus on WNT3A because of its high dysregulation in HNC, in addition to its correlation with cancer metastasis, which coincided with our functional results.

WNT3A is an upstream ligand of the well-known canonical Wnt signaling pathway. This pathway is observed across many types of cancers and affects different cancer functions, including cancer stemness and metastasis [72]. In agreement with WNT3A’s mRNA expression, our results showed that its protein expression levels were also decreased in the presence of miR-503, and luciferase reporter assay confirmed that miR-503 directly interacts with WNT3A on a translational level (Figure 3e,f). Unfortunately, the general interaction between miR-503 and WNT3A is not well known, with only one study by Tian et al. indicating their activity in leukemia [73]. Since WNT3A is an important ligand of the Wnt signaling pathway, we investigated whether miR-503 was correlated with this downstream cascade.

The Wnt signaling pathway is a well-known intercellular cascade that is tightly associated with cancer. The pathway is highly conserved in animals, meaning that the functions and associated genes are similar across animal species [74]. Most notably, highly conserved WNT proteins, approximately 40 kDa in size, were implicated in multiple functions, including stem cell control, embryogenesis, cell proliferation, and metastasis [2,75]. Moreover, WNT3A, a well-known prototypical Wnt ligand, is also known to play an important role in cancer metastasis through the Wnt signaling pathway by regulating extracellular matrix adhesion and invasion [76,77]. Specifically, WNT3A was reported in many cancer studies as a major participant in malignant cell invasion, including cancer types such as hepatocellular carcinoma [78,79,80], glioma [81], breast cancer [82], esophageal cancer [83], and colon cancer [84]. Genes and gene families, such as extracellular matrix genes, MMPs, c-Myc, and cyclin D1, are all important genes regulated by this pathway [85]. While there are various reports regarding the Wnt signaling pathway and HNC [86,87,88], to our knowledge, evidence regarding the WNT3A ligand and its relationship with miRNAs and HNC is yet to be determined. Therefore, we further investigated whether miR-503 affected the genes in the Wnt signaling pathway.

PCR arrays are an effective way to quickly analyze a large number of specific genes, such as those from a particular signaling pathway. This allowed us to specifically and efficiently screen for target genes from a large, organized gene bank [89]. In this study, a PCR array confirmed multiple associated Wnt targets besides WNT3A, including MMP7, FGF7, and CTGF, all of which are common extracellular matrix molecules (Figure 4a). FGF7 and CTGF were previously reported to play roles in multiple carcinogenic functions, such as cell proliferation, migration, and invasion [90,91]. Importantly, increased MMP regulation was frequently observed during cell invasion [92], and MMP7 was observed to be downregulated along with WNT3A in our PCR results. The interactions between MMPs and Wnt ligands have been extensively studied. Since the Wnt transduction pathway is frequently associated with tumor progression, and protease influx from the invading cells through MMP regulation is an important mediator of cell invasion, we proposed that the regulation of MMP molecules, such as MMP7, may occur in response to WNT3A and Wnt signaling pathway stimulation.

Finally, we analyzed invasion-specific genes to confirm the function of miR-503. The extracellular matrix pathway is closely associated with cancer cell invasion [93]. Molecules outside the cell typically form a web-like matrix that provides the general structure of the cell. However, cancer cells invade this membrane by breaking through the extracellular matrix via various mechanisms [94]. Our RT-qPCR invasion study confirmed that common genes found in the extracellular matrix pathway, specifically the MMP family, were indeed regulated by miR-503. MMP3, 7, and 9, among other genes from the same family, were studied extensively and were found to play roles in a variety of cancers and diseases through various pathways [95,96]; they have even been used as anti-cancer drugs [97]. Moreover, MMP7, a member of the MMP family, was not only highlighted as a significantly downregulated gene in the PCR array results but was also found to be significantly dysregulated across all three HNC cell lines in our gene expression study. Therefore, we believe that cancer invasion may be one of the main functions of miR-503 in HNC and is inhibited through the regulation of WNT3A. However, further experimentation regarding the impact of WNT3A on invasion is required to validate these results. Taken together, our results suggest that miR-503 inhibits HNC invasion through the Wnt signaling pathway by targeting WNT3A, which inhibits its downstream invasion-associated molecules, such as MMP7. By completely mapping the network and pathways of cancer development in HNC, we may be able to provide new information that could prove to be beneficial for other related studies. In addition, the associated genes and molecules may be critical for the advancement of diagnosis, prognosis, and even treatment of patients with HNC.

## 4. Materials and Methods

### 4.1. Cells and Cell Cultures

Seven HNC cell lines (SAS, OECM1, FaDu, SCC25, OC3, CGHNC8, and CGHNC9) and five normal keratinocyte cell lines (CGHNK2, CGHNK4, CGK1, CGK5, and CGK6) were used. The cells were maintained as previously described [98]. Briefly, SAS, CGHNC8, and CGHNC9 cells were cultured in DMEM culture medium (Gibco^®^, New York, NY, USA), SCC25 cells were maintained in DMEM/F12 medium (Gibco^®^, New York, NY, USA), OECM1 cells were maintained in RPMI-1640 culture medium (Gibco^®^, New York, NY, USA), OC3 was maintained in 1:2 DMEM/ Keratinocyte Serum-Free Medium (Gibco^®^, New York, NY, USA), and FaDu cells were grown in MEM culture medium (Gibco^®^, New York, NY, USA). All medium contained 7% (*v*/*v*) fetal bovine serum, 1% (*v*/*v*) antibiotic-antimycotic (100 mg/mL penicillin, 10 mg/mL streptomycin, and 25 g/mL amphotericin B) (Corning, Corning, NY, USA). The normal keratinocyte cell lines were cultured in KSFM (Gibco^®^, New York, NY, USA). The cultures were incubated at 37 °C in a humidified atmosphere containing 5% CO_2_-95% air.

### 4.2. miRNA-503 Mimic Construction and Transfection

Human miR-503 mimics were constructed and purchased from Shanghai GenePharma Co., Ltd. Forward 5′-UAGCAGCGGGAACAGUUCUGCAG-3′ and reverse 5′-UUAUCGUCGCCCUUGUCAAGACG-3′ miR-503 sequences were used for the construction of the mimic. The cells were plated at a density of 1 × 10^6^ in a 10 cm^2^ dish and cultured for 16 h. They were then transfected using Lipofectamine^®^ 2000 (Invitrogen™, Waltham, MA, USA) or the Jetprime^®^ transfection reagent (Polyplus^®^, Berkeley, CA, USA). The transfections were performed according to the protocol provided by the manufacturer. The RT-qPCR analysis was performed to determine the efficacy of the transfection. The Wnt agonist treatment (cat. no. sc-222416a; Santa Cruz Biotechnology, Inc., Dallas, TX, USA) in cells was performed with a dose of 5 ng/mL for 24 h at 37 °C prior to use.

### 4.3. RT-qPCR Analysis for miRNA-503 and Target Gene Expression

Cell pellets were collected from the cells transfected with the miR-503 mimics. The total RNA extraction was performed using the TRIzol reagent (Gibco BRL). Plasma RNA was harvested from 46 patient samples and converted to cDNA through reverse transcription. Taqman primer/probes for the plasma analysis were designed with sequences obtained from miRBase, and manufactured by Tri-I Biotech. The RT-qPCR analysis was performed as previously described. Briefly, miR-503-specific stem-loop RT primers were used for reverse transcription, and TaqMan miRNA assay kits (ABI) were used to measure the miR-503 expression. The results were normalized against a U6 internal control. iQ™ SYBR^®^ Green (Bio-rad, Hercules, CA, USA) dye was used for the analysis of the candidate genes. Primers were constructed by Tri-I Biotech, Inc., and were based on PrimerBank sequences. The experiments were performed in triplicates, and the statistical analysis was performed using Welch’s *t*-test.

### 4.4. Cell Proliferation Assay for Cell Growth Analysis

1 × 10^5^ to 5 × 10^5^ transfected cells were reseeded into 6-well plates and counted every 24 h for three consecutive days. The cell count for the miR-503 transfected cells was compared with the control counterparts.

### 4.5. Colony Formation Assay for Cell Growth Analysis

A total of 500 to 1000 transfected cells were seeded in 6-well plates and were allowed to grow for 10 to 14 days without disruption. The cells were then fixed and stained with crystal violet, and the cell colonies were counted. The experiments were conducted in triplicates, and the statistical analysis was performed using Welch’s *t*-test.

### 4.6. Wound-Healing Migration Assay

The cell migration was conducted using ibidi^®^ culture inserts (Applied BioPhysics, Inc., New York, NY, USA) placed in a 6-well plate. A total of 70, 80, and 90 µL of 5 × 10^5^ cells/mL transfectants were seeded into the inserts. After 16 h of incubation, the insert was detached, leaving a cell-free gap. A total of 2 mL of the appropriate medium with 0.5–1% FBS for each cell line was then added, and the cell migration status was photographed every 4 h, up to 16 h.

### 4.7. Matrigel Transwell Invasion Assay

The cell invasion was performed using Millicell^®^ (Millipore, Burlington, MA, USA) cell culture inserts. BioCoat Matrigel (Becton Dickinson Biosciences, Franklin Lakes, NJ, USA) was used to coat the transwell chambers for 6 h at 37 °C. The transfected cells were then seeded into the upper chamber at a density of 2 × 10^5^ cells. The lower chamber was then filled with 800 µL of regular culture medium with 20% FBS to promote cell invasion. The cells were incubated at 37 °C for 24 to 30 h. The invaded cells were fixed with formaldehyde, stained with crystal violet, and the number of invaded cells which passed through the Matrigel-coated membranes was counted and compared to their control chamber counterparts. The experiments were performed in triplicates, and the statistical analysis was performed using Welch’s *t*-test.

### 4.8. Western Blot Analysis of WNT3A Protein Expression

The total proteins were extracted using the CHAPS lysis buffer (10 mM Tris-base, pH 7.4, 1 mM MgCl_2_, 1 mM EGTA, 150 mM NaCl, 0.5% CHAPS, 10% glycerol, 10 mM Na_3_VO_4_, 10 mM NaF, and 1% (*v*/*v*) protease inhibitor), and incubated for 30 min on ice. The supernatant was harvested for protein quantification after centrifuging. The concentrations of the proteins were determined using the Bradford assay (Bio-Rad, Hercules, CA, USA). A 10% SDS-PAGE setup was used to separate the proteins, and this was transferred to a nitrocellulose membrane. Blocking was achieved with 5% milk and incubated overnight with rabbit WNT3A (GeneTex, Hsinchu, Taiwan; 1:1000) and mouse GAPDH (Merck Millipore, Burlington, MA, USA; 1:10,000) primary antibodies. The detection was performed using secondary anti-rabbit (Cell Signaling, Danvers, MA, USA; 1:5000) and anti-mouse (Cell Signaling, Danvers, MA, USA; 1:10,000) antibodies linked with horseradish peroxidase and enhanced chemoluminescence.

### 4.9. Bioinformatic Algorithm for the Target Gene Prediction of miRNA-503

The target prediction of miR-503 was performed using five online prediction algorithms: TargetScan version 7.2 (https://www.targetscan.org/vert_72/, accessed on 2 May 2021) [63], miRSystem version 20160513 (http://mirsystem.cgm.ntu.edu.tw/index.php, accessed on 18 September 2016) [64], DIANA (http://www.microrna.gr/microT-CDS, accessed on 1 September 2022) [65], miRanda (www.microrna.org, accessed on 1 March 2016) [66], and starBase version 2.0 (https://starbase.sysu.edu.cn/starbase2/, accessed on 1 September 2021) [67]. The common genes were grouped and depicted with jvenn software (http://jvenn.toulouse.inra.fr/app/index.html, accessed on 10 March 2022) [99]. 

### 4.10. Dual-Luciferase Reporter Assay to Confirm WNT3A Binding with miRNA-503

The luciferase reporter assay was performed as previously described [23]. Briefly, wild-type WNT3A 3′UTR and a mutant sequence were cloned downstream of the luciferase vector. The resulting plasmid was then co-transfected with miR-503. The Firefly and Renilla luciferase activities were measured using the Dual-Luciferase Reporter Assay System (Promega, Madison, WI, USA), according to the manufacturer’s instructions.

### 4.11. KEGG Enrichment Analysis to Examine the miRNA-503 Regulation Pathways

The KEGG enrichment of the genes predicted through the online databases was examined and interpreted using the DAVID database (https://david.ncifcrf.gov/, version 6.8, accessed on 1 April 2022). The KEGG data were extracted and transformed into a heatmap visualization, which was plotted with https://www.bioinformatics.com.cn/en, accessed on 23 April 2022, a free online platform for data analysis and visualization.

### 4.12. PCR Array Profiling of the WNT Signaling Target Genes

The profiling of genes was performed using Qiagen’s PCR array kit, according to the manufacturer’s protocol. Briefly, FaDu cells were transfected with miR-503, and the RNA was extracted and quantified as previously described. The cDNA synthesis was performed using Qiagen’s RT^2^ First Strand Kit (Cat. No. 330401) and combined with the RT^2^ SYBR^®^ Green PCR master mix (Cat. No. 330504). The master mix was then added to the Qiagen’s RT² Profiler™ PCR Array Human WNT Signaling Targets (Cat. No. PAHS-243Z) PCR array and the GeneGlobe Data Analysis Center was used to quantify the results. A cut-off value of |FR| ≥ 2 was set to differentiate between the significantly dysregulated genes.

## 5. Conclusions

In summary, we found that miRNA-503 acts as a tumor suppressor in HNC by inhibiting cell invasion through the suppression of WNT3A (Figure 5). As Wnt ligands activate the Wnt signaling pathway, we believe that miR-503 may directly regulate this pathway. Furthermore, invasion-associated genes downstream of the Wnt signaling pathway, especially several members of the MMP family, were also found to be dysregulated, leading us to conclude that miR-503 acts through the Wnt signaling pathway via the WNT3A/MMPs axis to regulate invasion in HNC. 

## Figures and Tables

**Figure 1 ijms-23-15900-f001:**
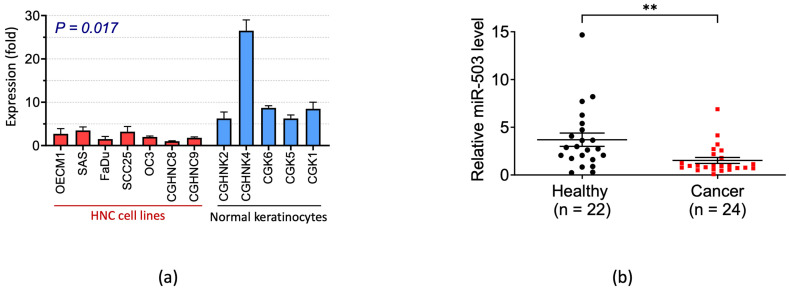
Differential levels of miRNA-503 between normal subjects and HNC. (**a**) The miR-503 expression was significantly decreased in HNC cell lines. HNC cell lines (*n* = 7) and normal oral keratinocytes (*n* = 5) were used to examine the miR-503 expression levels using RT-qPCR. (**b**) Cancer patients had significantly lower levels of miR-503 in the plasma. The miR-503 levels were determined using plasma samples from HNC patients (*n* = 24) and healthy individuals (*n* = 22). The statistical significance was calculated with Welch’s *t*-test (** *p* = 0.0055).

**Figure 2 ijms-23-15900-f002:**
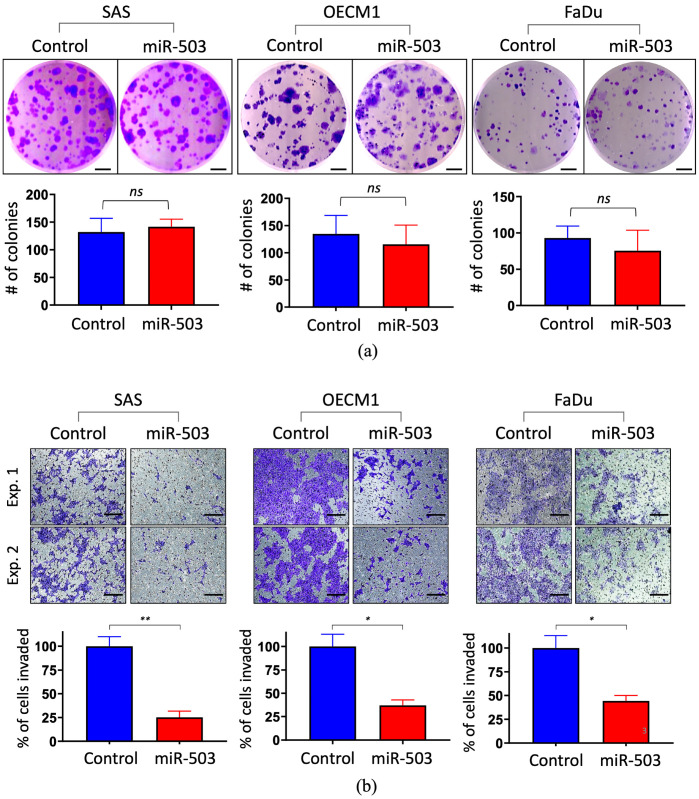
Functional analysis of miRNA-503 in HNC cells. The HNC cells (SAS, OECM1, and FaDu) were subjected to a functional analysis after transfection with a miR-503 mimic. (**a**) The colony formation assay showed no significant growth ability across all three cell lines, scale bar 5 mm. (**b**) The Matrigel invasion assay was used to determine the invasion ability, scale bar 100 μm. All functional assays were performed in triplicates. (** *p* ≤ 0.01, * *p* ≤ 0.05, *t*-test, ns = not significant).

**Figure 3 ijms-23-15900-f003:**
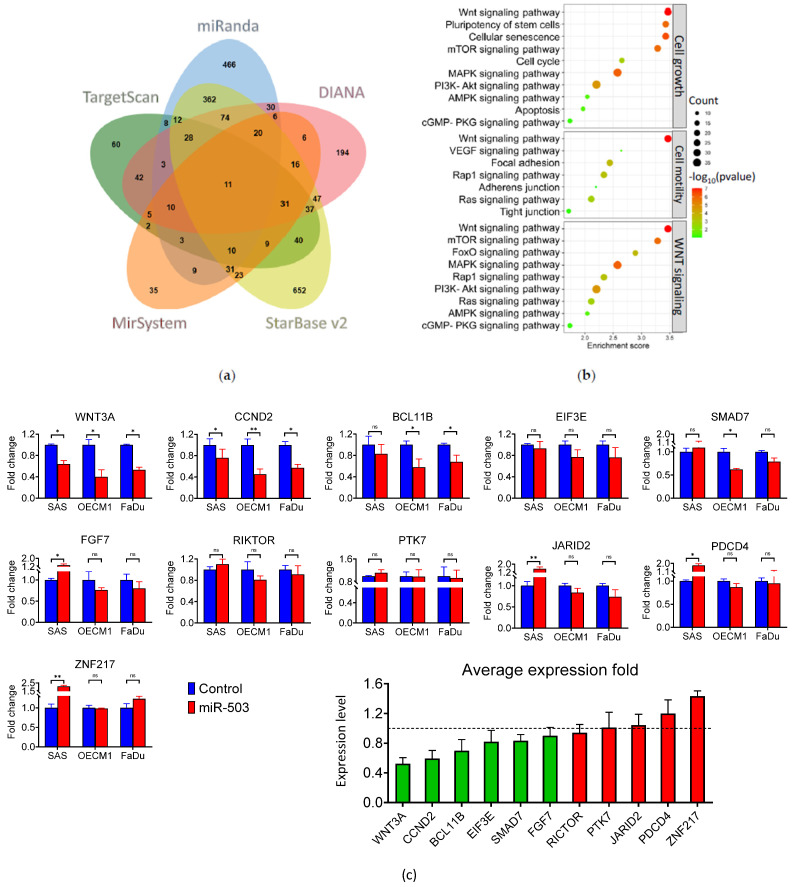
Prediction and validation of WNT3A as a target of miRNA-503. (**a**) Bioinformatics approach for the prediction of miR-503 genes. The TargetScan, miRanda, DIANA, miRSystem, and starBase prediction algorithm databases were incorporated. (**b**) The biological processes of the predicted genes from the five databases were analyzed with KEGG. The most prominent functions included cell growth, cell motility, and WNT signaling. (**c**) The RT-qPCR analysis of the 11 commonly predicted genes of miR-503. The experiments were performed with the SAS, OECM1, and FaDu cell lines and the average expression is shown (**c**, bottom). (**d**) The Western blot analysis showed that the WNT3A protein levels were significantly inhibited when miR-503 was overexpressed in SAS, OECM1, and FaDu cells. All the experiments were performed at least three times independently, and a typical result was shown. The error bars shown in the relevant figures indicate the standard deviation of the quantification results. GAPDH was used as an internal control to normalize the relative density levels of WNT3A. (**e**) A schematic representation of the luciferase reporter construction. The wild-type and mutant forms of the WNT3A 3′ UTR were co-transfected with a miR-503 plasmid for the dual-luciferase reporter assay. (**f**) The results were normalized to the mutant form luciferase activity levels. WNT3A co-transfection with miR-503 significantly inhibited the luciferase activity. (**g**) A rescue experiment restored the invasive ability of HNC cells when miR-503 was co-transfected with a Wnt agonist, scale bar 100 μm. The experiments were performed in triplicates. (** *p* ≤ 0.01, * *p* ≤ 0.05, *t*-test, ns = not significant).

**Figure 4 ijms-23-15900-f004:**
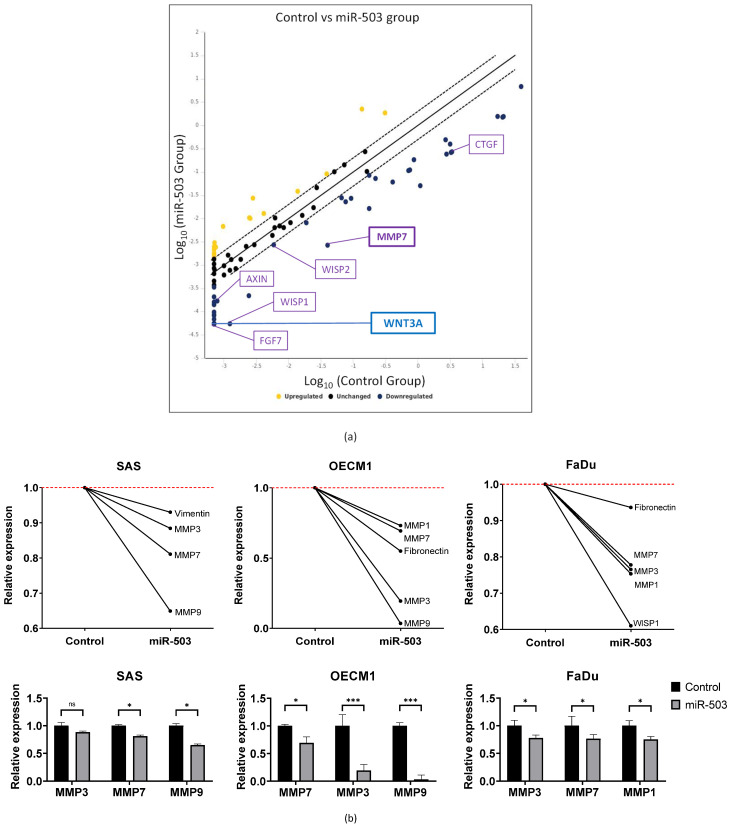
MiRNA-503 inhibits cell invasion through the WNT signaling pathway. (**a**) A PCR array was used to confirm the expression of WNT3A, as well as to search for targets associated with the WNT signaling pathway that may be deregulated by miR-503. (**a**) Scatter plot of the 84 genes in the array. An |FR| ≥ 2 threshold was used to separate the upregulated and downregulated genes. WNT3A was significantly downregulated, along with other important invasion genes such as MMP7, FGF7, and CTGF. (**b**) To determine the impact of invasion in HNC, a select list of genes commonly associated with cell motility was analyzed with RT-qPCR. Specifically, the expression levels of MMP family molecules were inhibited in the presence of miR-503, confirming the invasive function of miR-503 in HNC cells.The experiments were performed in triplicates. (* *p* ≤ 0.05, *** *p* ≤ 0.001, *t*-test, ns = not significant).

**Figure 5 ijms-23-15900-f005:**
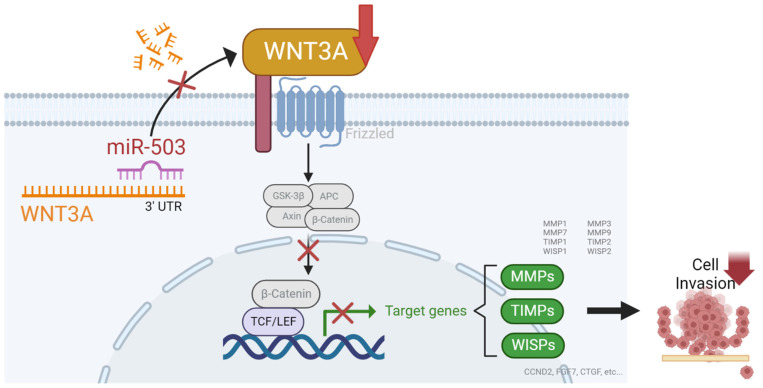
Summary figure of the miRNA-503/WNT3A pathway in HNC. Normally, activated WNT signaling induces β-Catenin translocation to the nucleus, promoting the transcription of MMPs, TIMPS, WISPs, and other invasion-associated genes. However, the overexpression of miR-503 subdues the activity of the WNT signaling pathway through WNT3A suppression, decreasing transcriptional activity, which results in the inhibition of HNC invasion.

## Data Availability

The data presented in this study are available upon request from the corresponding author.

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
