# Peer review of "Tumor Suppressor miRNA-503 Inhibits Cell Invasion in Head and Neck Cancer through the Wnt Signaling Pathway via the WNT3A/MMP Molecular Axis"

_ijms, 2022, doi:10.3390/ijms232415900_

Round 1

Reviewer 1 Report

Dear Authors,

In this paper, you provided evidence about the role of miR-503 as a tumor suppressor in HNC, inhibiting cell invasion through the Wnt signaling pathway via the WNT3A/MMP molecular axis.

The reviewer has read with interest this study. The paper is well written, and the experiments are adequate.

However, there are some questions for the authors:

1)     The functional analysis of miR-503 in HNC cells was performed in SAS, OECM1, and FaDu cells lines. However, the expression level of miR-503 (Fig.1A) in FaDu cell is not reported. Why the authors decided to use also the FaDu cells for the functional experiments?

2)     The functional analysis of miR-503 in HNC cells was tested by cell growth and cell invasion assays, using a colony formation assay and matrigel invasion assay. However, the functional analysis could be implemented using also a senescence assay (which is one of the prevalent pathway revealed by KEGG analysis in Fig.3B), and a migration assay like cell scratch assay to obtained a more detailed functional analysis of miR-503 in in HNC cells. 

3)     The authors shows that cell invasion (Fig.2B) was suppressed in HNC cell lines by miR-503 over-expression. Moreover, miR-503 modulates WNT3A at both the protein and mRNA levels (Fig.3C). The authors suggest that miR-503 modulates invasion through the WNT3A downstream signaling molecules. However, to confirm that WNT3A downregulation is directly involved in cell invasion suppression, a cell invasion assay after WNT3A downregulation could be performed and compared with miR-503 over-expression.

4)     In Fig.3C the error bar and the relative statistic are missing. Please provide additional informations. The ZNF217 graph is not formatted as the others presented in the same figure. Please change the graph. The average expression graph (3C bottom) is not clear:
- The average was calculated in miR-503 transfected OECM1 and FaDu cells?
- For PCD4 expression level, the average expression graph showed an expression level greater than 1.2 fold. However, the relative PCD4 graph showed a fold change close to 1 for both OECM1 and FaDu treated cells. How is possible? The same is for 
PTK7, JARID2, RIKTOR and ZNF217

5)     In Fig.3F, the luciferase assay was performed using wild type and mutant forms of WNT3A 3’ UTR co-transfected with miR-503 plasmid. However, the graph is not clear because in the x-axis was reported mutant vs miR-503, but it could be mutant vs wt. Please change the x-axis names.

6)     In Fig.4A-B the error bar, the relative statistic, and the information about biological experimental repeats are missing. Please provide additional informations.

7)     Lines 374-375, please write correctly.

8)     Line 396, please correct 2x105 cells with 2x105 cells

Kind regards.

Author Response

Response to Reviewer 1 Comments

Point 1: The functional analysis of miR-503 in HNC cells was performed in SAS, OECM1, and FaDu cells lines. However, the expression level of miR-503 (Fig.1A) in FaDu cell is not reported. Why the authors decided to use also the FaDu cells for the functional experiments?

Response 1: Thank you for your question. For the functional experiements, we included FaDu cells due to its high transfection efficiency for miR-503. We have revised Figure 1A, and have included the expression level of miR-503 in FaDu cell lines.

Point 2: The functional analysis of miR-503 in HNC cells was tested by cell growth and cell invasion assays, using a colony formation assay and matrigel invasion assay. However, the functional analysis could be implemented using also a senescence assay (which is one of the prevalent pathway revealed by KEGG analysis in Fig.3B), and a migration assay like cell scratch assay to obtained a more detailed functional analysis of miR-503 in in HNC cells.

Response 2: Thank you for your suggestion. We have performed a wound-healing migration assay for the functional analysis of miR-503 in HNC cells, and have included it in our supplementary data. We found that the upregulation of miR-503 in HNC cells did not have a notable effect on cell migration, and have noted this in the main text, on lines 135-137, stating “… Wound-healing migration assay was also performed, though no notable effect was seen when miR-503 was upregulated in HNC cells (Supplementary Figure S2B)…”.

Regarding senescence, our cell proliferation assay showed no significant difference in growth rate after miR-503 modulation (Supplementary Figure S1). Since a slower growth rate often accompanies cellular senescence, the insignificant effect of cell proliferation by miR-503 knockdown implies that miR-503 may have a minimal role in the senescence regulation. Nevertheless, further experiments may be needed to confirm this postulation. We have also added this point in the discussion section in lines 300-305.

Point 3: The authors shows that cell invasion (Fig.2B) was suppressed in HNC cell lines by miR-503 over-expression. Moreover, miR-503 modulates WNT3A at both the protein and mRNA levels (Fig.3C). The authors suggest that miR-503 modulates invasion through the WNT3A downstream signaling molecules. However, to confirm that WNT3A downregulation is directly involved in cell invasion suppression, a cell invasion assay after WNT3A downregulation could be performed and compared with miR-503 over-expression.

Response 3: Thank you for your suggestion. We acknowledge the importance of rescue experiments to strengthen our results, and would like to explore these findings in the future. However, due to the tine restraints for preparing a WNT3A knockdown/knockout model, obtaining these results may not be currently feasible. Alternatively, we have performed a literature search and compiled a few studies which support the finding that WNT3A is involved with cell invasion in our discussion section, from lines 341-344 in the revised manuscript. Additionally, we have also outlined this limitation in the discussion section on lines 378-380, stating that “… further experimentation regarding the impact of WNT3A on invasion is required to validate these results ” in the revised manuscript.

Point 4: In Fig.3C the error bar and the relative statistic are missing. Please provide additional informations. The ZNF217 graph is not formatted as the others presented in the same figure. Please change the graph. The average expression graph (3C bottom) is not clear:

- The average was calculated in miR-503 transfected OECM1 and FaDu cells?

- For PCD4 expression level, the average eexpression graph showed an expression level greater than 1.2 fold. However, the relative PCD4 graph showed a fold change close to 1 for both OECM1 and FaDu treated cells. How is possible? The same is for PTK7, JARID2, RIKTOR and ZNF217.

Response 4: Thank you for your question. Initially, we mistakenly excluded SAS in the individual bar graphs, so the average results look inconsistent. We have now added the SAS data to our individual graphs in Figure 3C, and fixed the errors regarding graph size and statistics.

Point 5: In Fig.3F, the luciferase assay was performed using wild type and mutant forms of WNT3A 3’ UTR co-transfected with miR-503 plasmid. However, the graph is not clear because in the x-axis was reported mutant vs miR-503, but it could be mutant vs wt. Please change the x-axis names.

Response 5: Thank you for your feedback. We have changed the x-axis of the graphs in Figure 3F from mutant vs. miR-503 to mutant vs wt.

Point 6: In Fig.4A-B the error bar, the relative statistic, and the information about biological experimental repeats are missing. Please provide additional informations.

Response 6: Thank you for your feedback. For Figure 4A, since the experiment is of an array nature, there were no experimental repeats, and therefore there are no available error bars and statistics. Figure 4B acts as a representative figure to emphasize the genes downregulated by miR-503, specifically for MMP family members. We have added an additional figure to Figure 4B (bottom), which shows the statistical significance of these genes.

Point 7: Lines 374-375, please write correctly.

Response 7: Thank you for your feedback. We have corrected lines 414-416 to “Cell pellets were collected from cells transfected with miR-503 mimics. Total RNA extraction was performed using TRIzol reagent (Gibco BRL).…” in the revised manuscript.

Point 8: Line 396, please correct 2x105 cells with 2x105 cells

Response 8: Thank you for your feedback. We have corrected this mistake, and have changed line 437 to “ … 2x105 cells …” in the revised manuscript.

Reviewer 2 Report

In this article, Tang et al identify miR-503 as a potential tumour suppressor gene in HNC. They propose that mir-503 prevents cell invasion by downregulating different target genes of the Wnt signalling pathway. Overall, further experiments would be needed to support the conclusions reached.

Major points:

-       Proliferation or MTT assays are hugely encouraged here to support the hypothesis that mir-503 is not involved in HNC cell proliferation. Colony formation assays are typically used to measure the oncogenic/transformation ability of cancer cells by evaluating their capacity to form colonies.

-       -In fig 3c, quantifications of the western-blot do not seem very reliable. To me the biggest differences in the amount of WNT3A protein levels between conditions are seen in FaDu cells. However, in SAS cell line the differences seem almost insignificant. This does not correspond with the quantification shown in the bar plot on the right. qPCR of Wnt3a should be performed to support the wb.

-       To proof that WNT3A and its targets are responsible for the reduced invasion seen after mir-503 overexpression, rescue experiments should be performed where WNT3A is overexpressed together with mir-503. Invasive phenotype should in theory be reversed to control levels. Also, invasion assays after siRNA silencing of WNT3A should be performed to address this issue.

-       How is WNT3A mRNA expression in HNC cell lines? Is there a correlation with the levels of mir-503 shown in figure 1a?

Minor points.

-       -In general, the language used is quite vague. For instance, lines 58-59, the term “molecules” should be replaced by a more specific term. The same happens in lane 74 with the term “molecular activities”. In general, the term “modulates” is also very vague and can be replaced by more specific terms throughout the manuscript, like terms such as “increases/decreases”, etc. Another example is found in line 97 with the sentence “mir-503 may affect HNC invasion”. Instead of “affects”, “reduces”, “decreases”, or something along these lines is more informative. This should be reviewed throughout the whole manuscript.

-       Error bars in figure 3f are missing for the mutant group.

-       -In the section 2.1 of the results  miR-503 levels in plasma from HNC patients are increased. Why is this? This should be discussed.

Author Response

Response to Reviewer 2 Comments

Point 1: Proliferation or MTT assays are hugely encouraged here to support the hypothesis that mir-503 is not involved in HNC cell proliferation. Colony formation assays are typically used to measure the oncogenic/transformation ability of cancer cells by evaluating their capacity to form colonies.

Response 1: Thank you for your suggestion. We have performed cell proliferation assays for the functional analysis of miR-503 in HNC cells, and included them in our supplementary data (Supplementary Figure S1A). We found that the upregulation of miR-503 in HNC cells did not have a notable effect on cell growth, which supports our initial hypothesis. We have noted this in the main text, on lines 129-131, stating “… Additionally, cell proliferation assay showed no significant effects on cell growth either, further indicating that miR-503 does not affect HNC cell growth (Supplementary Figure S1A)...” in the revised manuscript.

Point 2: In fig 3c, quantifications of the western-blot do not seem very reliable. To me the biggest differences in the amount of WNT3A protein levels between conditions are seen in FaDu cells. However, in SAS cell line the differences seem almost insignificant. This does not correspond with the quantification shown in the bar plot on the right. qPCR of Wnt3a should be performed to support the wb.

Response 2: Thank you for your feedback. In Figure 3D, We have re-selected a more suitable representative  figure for SAS cells, and described this assay in more detail in the Figure legend. “For each sample, the actin level was used as an internal control. The relative density of each WNT3A level was determined after normalization to the actin level. All the experiments were performed three times independently, and a typical result was shown. The error bars shown in the relevant figures indicated the standard deviation of the quantification results.” Also, the RT-qPCR of Wnt3a was performed and shown in Figure 3C.

Point 3: To proof that WNT3A and its targets are responsible for the reduced invasion seen after mir-503 overexpression, rescue experiments should be performed where WNT3A is overexpressed together with mir-503. Invasive phenotype should in theory be reversed to control levels. Also, invasion assays after siRNA silencing of WNT3A should be performed to address this issue. 

Response 3: Thank you for your suggestion. We acknowledge the importance of rescue experiments to fully confirm our hypothesis. Due to time constraints for preparing and designing these cell models, we have alternatively modified some of our wording throughout the main text to better correlate with our findings. We have also provided more references concerning WNT3A’s effects on cancer cell invasion in the discussion section on lines 341-344. Additionally, we have also outlined this limitation in the discussion section on lines 378-380, stating that “… further experimentation regarding the impact of WNT3A on invasion is required to validate these results.”.

Point 4: How is WNT3A mRNA expression in HNC cell lines? Is there a correlation with the levels of mir-503 shown in figure 1a?

Response 4: Thank you for your question. We have analyzed the basal levels of WNT3A expression in six HNC cell lines (SAS, OECM1, FaDu, OC3, CGHNC8, and CGHNC9), as well as three normal keratinocytes (CGHNK2 and CGK5, and CGK6). We found that the levels of WNT3A and miR-503 are inversely correlated, as shown in Supplementary Figure S2. This supports our hypothesis that WNT3A may be a direct target of miR-503, and we have outlined this finding in the revised manuscript in lines 164-167.

Point 5: In general, the language used is quite vague. For instance, lines 58-59, the term “molecules” should be replaced by a more specific term. The same happens in lane 74 with the term “molecular activities”. In general, the term “modulates” is also very vague and can be replaced by more specific terms throughout the manuscript, like terms such as “increases/decreases”, etc. Another example is found in line 97 with the sentence “mir-503 may affect HNC invasion”. Instead of “affects”, “reduces”, “decreases”, or something along these lines is more informative. This should be reviewed throughout the whole manuscript.

Response 5: Thank you for your feedback. We have reviewed the entire article and made appropriate changes in regard to language clarity.

Point 6: Error bars in figure 3f are missing for the mutant group.

Response 6: Thank you for your feedback. We have added the error bars in Figure 3F for the mutant group.

Point 7: In the section 2.1 of the results miR-503 levels in plasma from HNC patients are increased. Why is this? This should be discussed.

Response 7: Thank you for your question. In section 2.1, Figure 1B, miR-503 levels in plasma from HNC patients (Fig. 1B right side, red) are overall lower than that of healthy patients (Fig. 1B left side, black). These results are consistent with our hypothesis that miR-503 plays a tumor suppressor role in HNC.

Reviewer 3 Report

The authors present a very interesting work in the context of microRNAs as molecules involved in the process of carcinogenesis; their experimental approach is strongly supported by a bioinformatics analysis and has an orderly and well laid out sequence. However, it presents several aspects that need to be resolved and which I mention below:

- It is important that the authors homogenize and refer properly under the correct nomenclature to designate the abbreviated mature miRNA product (miR-503), origin within the genome (mir-503 [italics]) and its formal long name (miRNA-503 or microRNA-503 [e.g. line 364]). This suggestion may seem superficial, but it speaks of a correct conceptualization of this molecule. In this sense in your section: Results, line 100, it should be "2.1 miRNA-503 (miR-503) is downregulated in..." and so on in the rest of the document.

- Results section, line 103, says "OE3" should read "OC3".

- The title is incorrect (Results section, line 100), the authors are not showing that "something" downregulates miR-503, and they are only showing that there is low expression, so it should be "downnexpressed".

- Results section, lines 99 to 116. The authors mention that they used 5 normal keratinocyte lines, while in their materials and methods section, they mention that they used 4 (lines 348-352). Please resolve this inconsistency.

- Results section, caption figure 1. By convention, I would suggest to the authors to write the difference that corresponds to two asterisks (**) at the end of the caption of figure 1 and not only mention it in the text.

- Results section, caption figure 1, page 3, lines 114-115. The authors mention that they worked with 22 healthy patients and 24 cancer patients, and in the materials and methods section (page 11, line 376) they mention that they worked with samples from 48 patients, the authors should resolve this inconsistency or clarify that they lost two samples.

- What is the justification for not using the SAS cell line in gene expression assays (figure 3C), when it had a greater response to miR-503 transfection in both matrigel migration assays and western blot analysis? If there is any, please provide it in the text.

- I know that in Figure 3A, the important data are the 11 crossover genes in the center of the analysis done by the five databases, however, there are inconsistencies with the distribution of the crossovers: the authors report in starBase 1411 genes (my confirmation 1403), TargetScan 311 (311), DIANA 563 (560), miRanda 1091 (1083) and miRSystem 227 (227). If the image is not accurate and is only illustrative, the authors should clarify this in the figure caption or delete it, if they wish to keep it, they should show a perfect match of the gene crosses.

- Discussion section, second paragraph, page 9, lines 234-236. I do not agree with the statement the authors state here: "...Herein, we have found that miR-503 acted as a tumor suppressor in both HNC cell lines and patients with HNC (Figure 1)..." The low expression (culture and cell) and low presence in plasma of miR-503 (plasma) is not a finding that confirms its function as a tumor suppressor, supports this proposal, but does not determine its function. In order to confirm this, it would be necessary to perform approaches (at least) with knockout or knockdown assays in in vivo models and confirm the development of tumors in the head and neck due to the absence of miR-503. Therefore, I ask the authors to rethink this idea from a humbler perspective.

- Discussion section, sixth paragraph, page 10, lines 292-294. I disagree with the authors' statement here: "...In agreement with WNT3A mRNA expression, our results showed that its protein expression levels also decreased in the presence of miR-503, and the luciferase reporter assay confirmed their direct binding to each other (Figure 3E, F)..." Their luciferase assay indicates that miR-503 is capable of interfering with WNT3A translation, but not of interacting with the protein. I believe that this causes a confusion that is embodied even in Figure 5, in which it appears to interact directly with the WNT3A ligand, when what should be happening is that there is less of the ligand in the extracellular medium. The authors should rethink Figure 5 since it is confusing and does not represent or summarize the findings of their work.

Minor observations

- Properly use the 2X105 superscript (line 396) as you have in other sections (line 36).

- In the discussion section first paragraph, the authors use square brackets to list their findings, and this is confusing; this is because the same sign is used for citation of references. I suggest that another sign be used, perhaps half parentheses.

Author Response

Response to Reviewer 3 Comments

Point 1: It is important that the authors homogenize and refer properly under the correct nomenclature to designate the abbreviated mature miRNA product (miR-503), origin within the genome (mir-503 [italics]) and its formal long name (miRNA-503 or microRNA-503 [e.g. line 364]). This suggestion may seem superficial, but it speaks of a correct conceptualization of this molecule. In this sense in your section: Results, line 100, it should be "2.1 miRNA-503 (miR-503) is downregulated in..." and so on in the rest of the document.

Response 1: Thank you for your suggestion. We have reviewed the entire article and have standardized the nomenclature for miRNA-503. We designated the formal name for titles and headings, while experimental data was uniformly named with the mature miRNA product nomenclature.

Point 2: Results section, line 103, says "OE3" should read "OC3".

Response 2: Thank you for your feedback. We have corrected this mistake in the article on lines 106-107 to “…Seven HNC cell lines (OECM1, SAS, FaDu, SCC25, OC3, CGHNC8, and CGHNC9)…” in the revised manuscript.

Point 3: The title is incorrect (Results section, line 100), the authors are not showing that "something" downregulates miR-503, and they are only showing that there is low expression, so it should be "downnexpressed".

Response 3: Thank you for your suggestion. We have made the appropriate correction on lines 103-104 to “MiRNA-503 is downexpressed in patients with HNC as well as HNC cell lines” in the revised manuscript.

Point 4: Results section, lines 99 to 116. The authors mention that they used 5 normal keratinocyte lines, while in their materials and methods section, they mention that they used 4 (lines 348-352). Please resolve this inconsistency.

Response 4: Thank you for your feedback. We have corrected this inconsistency, and have revised lines 391-392 to “… and five normal keratinocyte cell lines (CGHNK2, CGHNK4, CGK1, CGK5, and CGK6) were used” in the revised manuscript.

Point 5: Results section, caption figure 1. By convention, I would suggest to the authors to write the difference that corresponds to two asterisks (**) at the end of the caption of figure 1 and not only mention it in the text.

Response 5: Thank you for your suggestion. We have added the p-value that corresponds with two asterisks in figure caption 1, on line 122 in the revised manuscript. The revised caption was changed to “…Statistical significance was calculated with Welch’s t-test (**p = 0.0055).”.

Point 6: Results section, caption figure 1, page 3, lines 114-115. The authors mention that they worked with 22 healthy patients and 24 cancer patients, and in the materials and methods section (page 11, line 376) they mention that they worked with samples from 48 patients, the authors should resolve this inconsistency or clarify that they lost two samples.

Response 6: Thank you for your feedback. We have resolved this inconsistency, and have changed the material and methods section, lines 416-417, to “Plasma RNA was harvested from 46 patient samples and converted to cDNA through reverse transcription” to match our results.

Point 7: What is the justification for not using the SAS cell line in gene expression assays (figure 3C), when it had a greater response to miR-503 transfection in both matrigel migration assays and western blot analysis? If there is any, please provide it in the text.

Response 7: Thank you for your question. We mistakenly excluded SAS in the individual bar graphs, and have now added results from the SAS cell line to our graphs in Figure 3C for consistency.

Point 8: I know that in Figure 3A, the important data are the 11 crossover genes in the center of the analysis done by the five databases, however, there are inconsistencies with the distribution of the crossovers: the authors report in starBase 1411 genes (my confirmation 1403), TargetScan 311 (311), DIANA 563 (560), miRanda 1091 (1083) and miRSystem 227 (227). If the image is not accurate and is only illustrative, the authors should clarify this in the figure caption or delete it, if they wish to keep it, they should show a perfect match of the gene crosses.

Response 8: Thank you for your feedback. Because some databases had results of duplicate genes, we initially excluded these genes for bioinformatic analysis. However, we overlooked these exclusions when constructing the Venn diagram. We have clarified these inconsistencies, and have changed the text on lines 149-150 to “… identified 1403 genes, while TargetScan, DIANA, miRanda, and miRSystem analyses identified 311, 560, 1083, and 227 genes, respectively”, to match the data in Figure 3A in the revised manuscript.

Point 9: Discussion section, second paragraph, page 9, lines 234-236. I do not agree with the statement the authors state here: "...Herein, we have found that miR-503 acted as a tumor suppressor in both HNC cell lines and patients with HNC (Figure 1)..." The low expression (culture and cell) and low presence in plasma of miR-503 (plasma) is not a finding that confirms its function as a tumor suppressor, supports this proposal, but does not determine its function. In order to confirm this, it would be necessary to perform approaches (at least) with knockout or knockdown assays in in vivo models and confirm the development of tumors in the head and neck due to the absence of miR-503. Therefore, I ask the authors to rethink this idea from a humbler perspective.

Response 9: Thank you for your feedback. We acknowledge the importance of in vivo experimentation to fully confirm our hypothesis. We have taken a humbler approach, and have reiterated our statement on lines 260-262 to “… Herein, we have found that miR-503 behaved like a tumor suppressor, for it was underexpressed in both HNC cell lines and patients with HNC (Figure 1)” in the revised manuscript.

Point 10: Discussion section, sixth paragraph, page 10, lines 292-294. I disagree with the authors' statement here: "...In agreement with WNT3A mRNA expression, our results showed that its protein expression levels also decreased in the presence of miR-503, and the luciferase reporter assay confirmed their direct binding to each other (Figure 3E, F)..." Their luciferase assay indicates that miR-503 is capable of interfering with WNT3A translation, but not of interacting with the protein. I believe that this causes a confusion that is embodied even in Figure 5, in which it appears to interact directly with the WNT3A ligand, when what should be happening is that there is less of the ligand in the extracellular medium. The authors should rethink Figure 5 since it is confusing and does not represent or summarize the findings of their work.

Response 10: Thank you for your feedback. We acknowledge that luciferase report assay only confirms activity on a translational level, and have clarified this idea in both the text as well as Figure 5. We have changed lines 326-330 to “… In agreement with WNT3A’s mRNA expression, our results showed that its protein expression levels were also decreased in the presence of miR-503, and luciferase reporter assay confirmed that miR-503 directly interacts with WNT3A on a translational level (Figure 3E, F)” in the revised manuscript. We have also redrawn Figure 5 to better represent this idea.

Point 11: Properly use the 2X105 superscript (line 396) as you have in other sections (line 36).

Response 11: Thank you for your feedback. We have corrected this mistake, and have changed line 437 to “ … 2x105 cells.”

Point 12: In the discussion section first paragraph, the authors use square brackets to list their findings, and this is confusing; this is because the same sign is used for citation of references. I suggest that another sign be used, perhaps half parentheses

Response 12: Thank you for your suggestion. We have changed the square brackets to parentheses for clarity, on lines 245-248, which now read: “… (1) … (2) … (3) … (4) …”

Round 2

Reviewer 1 Report

Dear Authors,

thank you for your replies, and for the improvement of your manuscript with the experiments required. 

Kind regards.

Author Response

Thank you for your approval.

Reviewer 2 Report

I would like to thank the authors for their effort to address my suggestions. I still think though that rescue experiments to validate that WNT3A is responsible for the reduced invasion observed after mir-503 overexpression is key for publication of this work. These experiments can easily be done in 1 week, max 2 weeks.

Also, the reason why mir-503 levels are low  in plasma of HNC patients is not  explained. I understand that mir-503 , as a TSG in HNC, his mRNA expression levels are expected to be low in these tumours. But, why is it also downregulated in plasma/blood cells of HNC patients? 

Author Response

Response to Reviewer 2

1. I would like to thank the authors for their effort to address my suggestions. I still think though that rescue experiments to validate that WNT3A is responsible for the reduced invasion observed after mir-503 overexpression is key for publication of this work. These experiments can easily be done in 1 week, max 2 weeks.

Thank you for your suggestion. We have performed the invasion rescue experiment to determine the functional effects of mir-503 after WNT3A overexpression, and we demonstrated the restoration of the invasive phenotype when overexpression of WNT3A was able to reverse the anti-tumor effects of mir-503 in HNC, as shown in Figure 3G. We have also noted this finding in the main text, on lines 179-182, stating that “Rescue experiments were performed to validate the role of WNT3A in HNC. Overexpression of miR-503 alongside a Wnt agonist demonstrated the restoration of the invasive phenotype when it reversed the anti-tumor effects of mir-503 in HNC, as shown in Figure 3G.” 

2. Also, the reason why mir-503 levels are low in plasma of HNC patients is not explained. I understand that mir-503, as a TSG in HNC, his mRNA expression levels are expected to be low in these tumours. But, why is it also downregulated in plasma/blood cells of HNC patients? 

Thank you for your question. We have addressed this issue in the introduction on lines 70-72 and 80-82 to clarify the role of circulating miRNAs, and have added related references in the main text. Studies have shown that normally, tumor suppressor miRNAs (such as miR-503) are secreted by healthy cells to produce a homeostatic environment [16,17]. However, when tumor cells become more prevalent due to cancer progression, the amount of tumor suppressor miRNAs secreted is decreased due to this change in phenotype [29]. This is why we examined miR-503 in circulation, and found that it is downregulated in both HNC cells as well as in the plasma of HNC patients.

Reviewer 3 Report

I consider that the authors have given an adequate response to my observations on their work.

Author Response

Thank you for your approval.

Round 3

Reviewer 2 Report

I appreciate the authors' effort to address my suggestions. I have no more comments.

Author Response

Thank you for your approval.